# The effect of population-based blood pressure screening on long-term cardiometabolic morbidity and mortality in Germany: A regression discontinuity analysis

Sara Pedron[1,2,3]*, Michael Hanselmann[1], Jacob Burns[1,4,5], Alexander Rich[2,4,5], Annette Peters[6], Margit Heier[6,7], Lars Schwettmann[2,8], Jacob H. Bor[9,10], Till Bärnighausen[11,12,13], Michael Laxy[1,2,3,14]

1 Professorship of Public Health & Prevention, Technical University of Munich, Munich, Germany, 2 Institute of Health Economics and Health Care Management, Helmholtz Zentrum München, German Research Center for Environmental Health (GmbH), Munich, Germany, 3 German Center for Diabetes Research (DZD), Munich-Neuherberg, Munich, Germany, 4 Institute for Medical Informatics, Biometry and Epidemiology (IBE), Chair for Public Health and Health Services Research, LMU Munich, Munich, Germany, 5 Pettenkofer School of Public Health, Munich, Germany, 6 Institute of Epidemiology, Helmholtz Zentrum München, German Research Center for Environmental Health (GmbH), Munich, Germany, 7 KORA Study Center Augsburg, University Hospital of Augsburg, Augsburg, Germany, 8 Department of Health Services Research, School of Medicine and Health Sciences, Carl von Ossietzky University of Oldenburg, Oldenburg, Germany, 9 Department of Global Health, Boston University School of Public Health, Boston, Massachusetts, United States of America, 10 Health Economics and Epidemiology Research Office, Department of Internal Medicine, School of Clinical Medicine, Faculty of Health Sciences, University of Witwatersrand, Johannesburg, South Africa, 11 Heidelberg Institute of Global Health (HIGH), Medical School, Heidelberg University, Heidelberg, Germany, 12 Harvard Center for Population and Development Studies, Cambridge, Massachusetts, United States of America, 13 Africa Health Research Institute (AHRI), Somkhele and Durban, South Africa, 14 Global Diabetes Research Center, Rollins School of Public Health, Emory University, Atlanta, Georgia, United States of America

* sara.pedron@tum.de

**Data Availability Statement:** All data used in this manuscript are available via project agreement from the KORA study (https://helmholtz-

## Abstract

### Background

Hypertension represents one of the major risk factors for cardiovascular morbidity and mortality globally. Early detection and treatment of this condition is vital to prevent complications. However, hypertension often goes undetected, and even if detected, not every patient receives adequate treatment. Identifying simple and effective interventions is therefore crucial to fight this problem and allow more patients to receive the treatment they need. Therefore, we aim at investigating the impact of a population-based blood pressure (BP) screening and the subsequent "low-threshold" information treatment on long-term cardiovascular disease (CVD) morbidity and mortality.

### Methods and findings

We examined the impact of a BP screening embedded in a population-based cohort study in Germany and subsequent personalized "light touch" information treatment, including a hypertension diagnosis and a recommendation to seek medical attention. We pooled four waves of

muenchen.managed-otrs.com/external). A full variable list and the analysis code is available online at osf.io/pwgfh.

**Funding:** The authors received no specific funding for this work.

**Competing interests:** I have read the journal's policy and the authors of this manuscript have the following competing interests: TB is editor-in-chief of PLOS Medicine.

**Abbreviations:** BMI, body mass index; BP, blood pressure; BPS, blood pressure score; CI, confidence interval; CVD, cardiovascular disease; GP, general practitioner; HIC, high-income country; HR, hazard ratio; ITT, intention-to-treat; IV, instrumental variable; KORA, Cooperative Health Research in the Augsburg Region (study); LATE, local average treatment effect; LMIC, low- and middle-income country; MONICA, Monitoring Trends and Determinants in Cardiovascular Disease (study); RCT, randomized controlled trial; RD, regression discontinuity; WHO, World Health Organization.

the KORA study, carried out between 1984 and 1996 ($N = 14{,}592$). Using a sharp multivariate regression discontinuity (RD) design, we estimated the impact of the information treatment on CVD mortality and morbidity over 16.9 years. Additionally, we investigated potential intermediate outcomes, such as hypertension awareness, BP, and behavior after 7 years.

No evidence of effect of BP screening was observed on CVD mortality (hazard ratio (HR) = 1.172 [95% confidence interval (CI): 0.725, 1.896]) or on any (fatal or nonfatal) long-term CVD event (HR = 1.022 [0.636, 1.641]) for individuals just above (versus below) the threshold for hypertension. Stratification for previous self-reported diagnosis of hypertension at baseline did not reveal any differential effect. The intermediate outcomes, including awareness of hypertension, were also unaffected by the information treatment. However, these results should be interpreted with caution since the analysis might not be sufficiently powered to detect a potential intervention effect.

## Conclusions

The study does not provide evidence of an effect of the assessed BP screening and subsequent information treatment on BP, health behavior, or long-term CVD mortality and morbidity. Future studies should consider larger datasets to detect possible effects and a shorter follow-up for the intermediate outcomes (i.e., BP and behavior) to detect short-, medium-, and long-term effects of the intervention along the causal pathway.

## Author summary

### Why was this study done?

- High blood pressure (BP), or hypertension, if untreated, increases the risk of severe health issues and death.

- This issue is relevant globally, affecting high-income countries as well as low- and middle-income countries, where the problem is even more serious.

- Understanding efficient ways to identify people with hypertension and encourage them to seek treatment is very important to improving the health of populations.

### What did the researchers do and find?

- We assessed whether a simple message, given in the form of a letter to individuals identified as having high BP, which motivated them to visit their general practitioner to discuss the high BP, led to a reduced risk of stroke, heart attack, and other forms of heart disease, or to changes in lifestyle.

- To do so, we used data for 14,592 individuals from a long-running study in southern Germany, which reports data on its participants from 1984 to 1985, 1989 to 1990, 1994 to 1995, and 1999 to 2001.

- To determine whether providing people with high BP information through this simple message improved cardiovascular health, we compared their health and behavior to the

health and behavior of individuals with very similar BP who did not receive the information over a period of 17 years.

- We did not find evidence that providing information to individuals with high BP led to improved health or behavior during the study. Specifically, individuals just above the threshold for hypertension (that received a diagnosis) were not less likely than individuals just below the threshold (that did not receive the diagnosis) of dying from CVD (HR = 1.172 [95% CI: 0.725, 1.896]) or of suffering any (fatal or nonfatal) long-term CVD event (HR = 1.022 [0.636, 1.641]).

**What do these findings mean?**

- The findings of our study could mean multiple different things: that the messaging and follow-up provided to people with high BP needs to be more intense if they are to be motivated to act; that health and behavior of these people did change over a short but not over a long time period; or that our study was too small to accurately detect an improvement in health and behavior due to the provided information.

- In the future, similar yet larger studies from Germany and elsewhere that measure changes in health and behavior over a short-, medium-, and long-term period would be helpful in addressing this question.

## Introduction

Hypertension is one of the most important and prevalent risk factors globally for cardiovascular morbidity and mortality. According to recent estimates, every year, 8.5 million deaths worldwide are attributable to hypertension [1]. Early detection and management of this condition, which typically consists of lifestyle changes and blood pressure (BP)-lowering drug treatment, are vital to ensure successful prevention of cardiovascular disease (CVD) [2]. However, multinational studies have shown that a large proportion of hypertension cases remains undetected and that even after detection, many individuals remain untreated [3]. In high-income countries (HICs), the proportion of individuals aware of their hypertensive condition is below 50%, with only 47% of individuals treated after diagnosis [3]. In low- and middle-income countries (LMICs), this problem is even more acute, with 41% to 44% aware of being hypertensive and 32% to 37% treated after diagnosis [3]. Effective and cost-effective population-wide prevention strategies are needed to detect individuals with hypertension and motivate them to seek treatment.

Population-based screening is typically a "light touch" intervention, in which individuals are informed of their hypertension status and encouraged to consult a physician for treatment. However, the available evidence has shown mixed results regarding the effect of simply communicating a new diagnosis on medium-term behavioral changes and health improvements, both in the case of hypertension [4–9] and other diseases [10–18]. Studies in China and South Africa found that hypertension screening and counseling reduced BP and improved lifestyle outcomes at 2 years [4–7]. However, studies conducted in the United States showed only modest changes in dietary patterns up to 2 years after hypertension diagnosis [8,9]. Differences might stem from the methodology used, the different context, and the different screening

considered. In any case, the effect of such "light touch" BP screening on medium-term behavior and health remains unclear. Furthermore, to our knowledge, no evidence exists on the long-term effects of such population-based screenings on mortality and morbidity.

This study evaluates the effect of a population-based BP screening intervention on long-term CVD morbidity and mortality in Germany. Like other studies in this context [4–7], we used a quasi-experimental approach to evaluate the causal effect of the screening using data from a population-based epidemiological study. Quasi experimental studies offer the possibility of deriving estimates of causal impact from observational data in substantially less time and using substantially fewer resources than RCTs [19], in a context where experimental studies would not be feasible due to ethical and economic reasons [20,21]. We hypothesized that the information given after a BP screening would have motivated patients to seek medical attention. In the context of routine care in Germany, this would likely have led to initiation of a series of further diagnostic, behavioral, and therapeutic measures to keep BP levels under control and to reduce long-term CVD morbidity and mortality.

## Methods

### Study population

We used data from the population-based KORA platform (Cooperative Health Research in the Augsburg Region). The base for the KORA study consists of three World Health Organization (WHO) MONICA (Monitoring Trends and Determinants in Cardiovascular Disease) studies (S1, S2, and S3) that were started during the 1980s to 1990s as a general effort of WHO to generate population-based epidemiological data in several areas in Europe [22]. The study conducted in the city of Augsburg was then carried on and further developed within the KORA study, which added morbidity registries and follow-up studies and included new cohorts.

Specifically, we pooled data from the three cross-sectional WHO MONICA surveys S1 (1984/1985; $n = 4,022$), S2 (1989/1990; $n = 4,940$), S3 (1994/1995; $n = 4,856$), and the subsequent KORA survey S4 (1999 to 2001; $n = 4,261$). For the secondary outcomes analysis, we used the follow-up studies to S3 and S4, namely, KORA-F3 (2004 to 2005, $n = 2,586$) and KORA-F4 (2006 to 2008, $n = 2,544$), respectively. Furthermore, we used the CVD mortality and morbidity follow-up data that were collected longitudinally for all participants of the four cohorts. Recruitment and data collection in all MONICA and KORA studies followed a very similar, nearly identical, protocol. Details of the KORA study can be found in Holle and colleagues' paper [22]. All KORA studies were carried out in accordance with ethical regulations at the time the studies were initiated. The KORA studies F3, S4, and F4 were approved by the Ethics Committee of the Bavarian Medical Association (Ethics number: F3 03097, S4 99186, F4 06068). Written informed consent was obtained by all study participants.

In each MONICA/KORA study, participants underwent several medical examinations, blood tests, a structured interview, and a self-administered questionnaire. For the present analysis, we excluded individuals who lacked data for all relevant variables or who had a history of CVD events at baseline (stroke or myocardial infarction). Additionally, we excluded those who were already taking BP-lowering drugs at baseline, measured by a computer-assisted drug recording procedure (see S1 Appendix for details). Additionally, for the secondary outcomes sample, we excluded individuals who lacked follow-up data and individuals who lacked data on one or more of the analyzed outcomes.

We imputed missing values in the outcomes and covariates using a comprehensive model with a predictive mean matching approach with 20 replications using the mice package in R [23]. Further details on the imputation model and diagnostics are available in S2 Appendix. In

a sensitivity analysis, we assessed the robustness of estimation results using the original sample without imputed values.

## Intervention

All participants in the KORA study received clinical screening for BP, body mass index (BMI), cholesterol, and other parameters (S3 Appendix). Approximately 2 weeks after the survey, all participants received a letter with the results of their clinical screening and encouragement to seek a doctor if values were above recommended thresholds. For this study, we focus on communication of the results of the BP screening, which we consider a "light touch" intervention since individuals were simply informed of their diagnosis and encouraged to consult a physician through their own initiative.

As such, the assessed intervention is an information treatment targeted at high-risk individuals, following a BP screening within a population-based epidemiological study [24]. The intervention represents a one-time occurrence outside of the usual healthcare context and is therefore different than regular screening interventions carried out as part of primary care.

BP was measured in all MONICA/KORA surveys following a standard procedure [25], which involved three sequential measurements, two of which were performed after at least 30 minutes rest. We extracted data on the value of the participants systolic BP and diastolic BP at the screening. In accordance with the value communicated to participants, we extracted data on the third BP measurement for participants of the S1 to S3 studies, and we computed the average of the second and third BP measurement for participants of the S4 study.

The exposure of interest was whether the participant's BP result was above the designated threshold for hypertension (diastolic BP $\geq$ 90 mm Hg or systolic BP $\geq$ 140 mm Hg). Whereas all participants were screened for hypertension and received communication on their results, only those with BP above the threshold received an additional prompt with their hypertension diagnosis and encouragement to seek further care (see S3 Appendix for details). We compare participants diagnosed with hypertension with those not diagnosed.

## Outcomes

The primary outcome we considered was the incidence of any fatal CVD event (stroke, myocardial infarction, coronary heart disease and other CVD events–ICD-9 codes: 390 to 459, 798) within up to 16.9 years of follow-up (i.e., the longest common follow-up across surveys). Death certificates were received from local health authorities to confirm these events.

Furthermore, we investigated the occurrence of any CVD event, including both information on mortality (fatal CVD event) and morbidity (nonfatal CVD event) in one outcome variable. The nonfatal incident CVD events were assessed through the population-based Augsburg Coronary Event Registry or by postal follow-up questionnaires and were validated using data from participants' hospital records and their attending physician [26–28].

As secondary outcomes, we analyzed systolic and diastolic BP, hypertension awareness, physical activity, smoking, and BMI. These outcomes were assumed to lie on the causal pathway between information treatment and CVD outcomes (*intermediate outcomes*) and were measured in the 7-year follow-up studies for the S3 and S4 studies (i.e., F3 and F4, respectively). No follow-up data were available for the studies S1 and S2. Previous hypertension diagnosis was measured by asking the question "Have you ever been diagnosed with hypertension?" ("Yes," No," "I don't know"). The original question in German was: "*Ist bei Ihnen jemals hoher Blutdruck festgestellt worden*?", which translates as "Have you ever been diagnosed with hypertension?", but also includes the possibility of a diagnosis from someone other than the physician (oneself, a relative, a pharmacist, etc.) because the agent is not

specified. Both current smoking status and physical activity levels were self-reported. Frequency of physical activity was collected in four categories ("No physical activity," "Irregularly about 1 hour per week," "Regularly about 1 hour per week," "Regularly more than 2 hours per week"). We dichotomized this variable, considering individuals performing physical activity regularly at least 1 hour per week as "high physical activity," and "low physical activity" otherwise. BMI was computed based on objectively measured height and weight.

## Study design and statistical analysis

We evaluated the impact of a hypertension diagnosis and encouragement letter on health outcomes using a regression discontinuity (RD) design. RD enables causal inference when an intervention is assigned based on a threshold rule. We compared KORA study participants who had baseline BP readings just below the threshold for hypertension (and therefore did not receive an information treatment comprising hypertension diagnosis and encouragement to seek care) (*control group*) with study participants who had baseline BP readings just above the threshold (and therefore received hypertension diagnosis and an encouragement prompt) (*intervention group*). Due to random variability in BP measurements, participants just above/below the threshold, apart from receiving the information treatment, can be assumed to be similar on both observed *and unobserved* baseline characteristics, as in a randomized trial [29,30]. This means that differences outside of the study context, such as differing access and standards of care, should also be balanced across intervention and control groups.

In the analysis, individuals who are just above/below the cutoff were selected using a *bandwidth*, which determines the span of the distribution around the cutoff in terms of the assignment variable to be considered in the analysis (e.g., a bandwidth of 5 mm Hg would imply considering individuals between 135 mm Hg and 145 mm Hg for systolic BP). The optimal bandwidth was computed by optimizing the trade-off between being in the closest proximity of the cutoff to improve comparability of observations, while at the same time ensuring that enough observations are considered in the analysis [29,30].

Hypertension is diagnosed based on a compound threshold rule including both systolic and diastolic BP. Participants with a diastolic BP $\geq$ 90 mm Hg *or* a systolic BP $\geq$ 140 mm Hg received the information that their BP was too high and advice that they should seek medical attention. Because a participant could be diagnosed due to elevated levels of either diastolic BP (*dBP*) or systolic BP (*sBP*), we combined the two values into a single assignment variable with a single threshold rule or binding score (*Blood Pressure Score—BPS*) [31]. First, we created the single running variable by centering the two distinct diastolic and systolic BP variables at the respective cutoffs and divided the values by the respective standard deviation so that diastolic BP and systolic BP would have a similar density for observations in a close proximity around the threshold (i.e., within a given *bandwidth* around the threshold). Second, to ensure that each individual contributed only one value, we selected the maximum value for each individual from the new running variable, while disregarding the lower value:

$$BPS = \max\left[ \left(\frac{dBP - 90}{sd(dBP)}\right); \left(\frac{sBP - 140}{sd(sBP)}\right) \right]$$

Based on this running variable, treatment was assigned to all individuals with a *BPS* $\geq$ 0. Individuals with a BPS < 0 were assigned no treatment and represent thus the control group.

This reformulation to a single running variable has a major advantage compared to the scenario with two running variables, one for diastolic BP and one for systolic BP. It allows estimating the effect of treatment in the context of a sharp RD design, where no individual below the cutoff received the intervention and all individuals above the cutoff received the

intervention. The sharp design allows the estimation of the intention-to-treat effect (ITT) at the threshold, i.e., for those individuals in a close proximity to the threshold [29]. Considering the two assignment variables separately would have led to the case that some individuals were above the respective threshold for one BP dimension and below the threshold for the other—i.e., such an individual would have been assigned to receive the intervention according to the first BP dimension, yet assigned to the control for the latter. This situation would have required the estimation of two separate fuzzy RD designs, yielding the estimation of a local average treatment effect (LATE), only valid for compliers and thus not generalizable to the whole population [29,30].

One potential threat to validity of the RD designs is the possibility that the assignment variable (in our case, BP as measured in the study center) might be manipulated, e.g., to gain access to treatment [29,30]. This would occur if, for example, the KORA staff member conducting the physical examinations falsely registered systolic or diastolic BP measurements that were just below the threshold. This is unlikely as the study-based clinical exam was not part of routine patient care, so there was little motivation to manipulate screening values. We tested this assumption by graphically inspecting the density plot of the assignment variable and its components for each sample utilized in the analysis.

Furthermore, RD designs yield causal inferences because participants are comparable on either side of the threshold [29,30]. Similar to a balance table in a randomized controlled trial (RCT), we can show similarity on baseline observables. We estimate RD models replacing the outcome variable with baseline characteristics in order to assess differences at the threshold. Furthermore, we inspected a graphical depiction of covariates around the threshold.

Another issue regards the fact that the threshold that was adopted to warn individuals about their hypertension (diastolic/systolic BP $\geq$ 90/140 mm Hg) is also the threshold that is widely adopted by physicians to define and treat hypertension in the everyday clinical practice. However, given a substantial random element in the individual BP measurements, which can fluctuate by several units within hours, this does not represent a violation of the exclusion restriction and therefore does not threaten the internal validity of our estimator [32]. We used a standard linear model to estimate the effect of treatment exploiting the sharp RD design that takes on the following form:

$$y = \beta_0 + \beta_1 x + \tau\; cutoff + \beta_2\; x\; \times\; cutoff + \beta_j X_j + \epsilon$$

where $y$ is the outcome and $x$ is a continuous assignment variable, in our case the BPS. The variable *cutoff* is a dummy that takes on the value one if the assignment variable is above the deterministic treatment threshold, or zero otherwise. Therefore, $\beta_0$ represents the intercept, $\beta_1$ the effect of the assignment variable on the outcomes before the cutoff, and $\beta_2$ the change of this effect above the cutoff. The coefficient $\tau$ represents the treatment effect of interest, estimated within the optimal bandwidth around the threshold. $X_j$ represents a vector of j covariates (sex, age, age squared, survey wave), and the vector $\beta_j$ represents the respective coefficients. $\varepsilon$ is the idiosyncratic error.

We estimated the effect of treatment on the primary outcomes using survival analysis in a Cox proportional hazard model. For the analysis of fatal CVD events, participants were censored at the first date of record of death due to CVD causes, death for other causes, or after 16.9 years, whichever came first. For the analysis of any CVD events, participants were censored at the first date of record of any (fatal or nonfatal) CVD event, death for other causes, or after 16.9 years, whichever came first. To verify the proportional hazards assumption, we plotted Schoenfeld residuals. For the secondary outcomes, we used the same model specification,

using linear estimation for continuous outcomes and a logit specification for dichotomous outcomes.

We computed the optimal bandwidth around the treatment threshold using the Imbens–Kalyanaraman method, with a rectangular kernel [33]. Although this method was developed for linear models, no better methodology exists to compute such bandwidth for Cox or logit models. Therefore, we employed this method for all models. Furthermore, we tested sensitivity of our results to different bandwidth choices.

In our main analysis, the sample used for the long-term mortality and morbidity analysis differs from the sample used for the follow-up behavioral intermediate outcomes. Loss to follow-up (i.e., individuals who were discarded from the analysis because of missing values at follow-up) may have influenced the results for intermediate outcomes if individuals who died or who were not healthy enough to participate were more likely to refuse participation. For mortality and morbidity outcomes, we imputed missing data in the cause of death and CVD events, so no loss to follow-up occurred. This might have caused a mismatch for the two results. However, we chose to analyze these samples as such, because it allowed us to assess the largest number of participants for all outcomes.

The information treatment might have had different effects on individuals depending on their awareness of their hypertension status (self-reported previously diagnosed and untreated hypertension). Individuals who had already been diagnosed with hypertension before might react differently from individuals with no previous hypertension. Therefore, in a series of additional analyses, we stratified the analyses of primary outcomes by the self-reported previously diagnosed hypertension at baseline, again determined by the question "Have you ever been diagnosed with hypertension?" ("Yes," "No," "I don't know").

Finally, we tested the sensitivity of results to different bandwidths. All analyses were carried out using R (Version 4.0.3). A full variable list and the analysis code are available on the Open Science Framework (osf.io/pwgfh). For transparent reporting, we followed and compiled the TREND checklist (see S1 Checklist) [34].

## Protocol and analysis plan availability

We did not develop an a priori study protocol for this study. We initially planned and conducted explorative analyses to determine the feasibility of employing an RD design to evaluate the population-based BP screening on a CVD events. In these analyses, we employed a fuzzy RD design using both systolic and diastolic BP as separate cutoffs. After determining that the evaluation was indeed feasible, we set out to conduct this full study. During planning consultations among the team at this stage, we made two main changes to our initial explorative analyses. Firstly, we decided to combine the two cutoffs into a single threshold, the BPS, and to consequently employ a sharp RD design, as described above. Secondly, to increase clinical relevance, we decided not only to assess the composite outcome of any CVD event as a primary outcome, but to also assess any fatal CVD event. During the study, we ensured that we made decisions based only on methodological considerations and that no decisions were based on the results of the analyses.

## Patient and public involvement

Since this study was based on secondary data from existing data sources, it was not possible to involve study participants in the design, conduct, and dissemination plans of our study. Due to time and resource constraints, we could not include members of the public in the planning of the current study.

## Results

The initial sample for the main analysis comprised 17,602 individuals. After excluding individuals lacking data for almost all variables ($n = 112$), participants with a history of CVD events at baseline (stroke or myocardial infarction, $n = 556$), and participants who were already taking antihypertensive medications at baseline ($n = 2,744$), the final sample used for the main analysis entailed 14,592 individuals. Since we imputed missing information on the cause of death ($n = 102$) and CVD events (myocardial infarction $n = 345$, stroke $n = 987$), no individual was lost to follow-up in this sample. A flowchart of sample construction is available in S1 Fig.

The initial sample for the secondary outcomes analysis included 9,117 individuals. Following exclusion for the abovementioned criteria (lack of all variables, history of CVD at baseline, and antihypertensive medication intake–$n = 1,758$), the sample consisted of 7,359 individuals. Of these, $n = 2,225$ were lacking follow-up data (loss to follow-up is equal to 30%), and $n = 49$ were lacking at least one outcome variable. Therefore, the final sample for the secondary outcomes analysis included 5,085 individuals. A flowchart of the secondary outcomes sample construction is available in S2 Fig. For further details on both samples construction and imputation, see S2 Appendix and S1, S2, and S3 Tables.

The final sample (Table 1) included individuals between 24 and 75 years of age at baseline (mean age: 46 years), half of which was female (50%). Within the 16.9 years we included in the analysis, we identified 637 (4.4%) individuals with fatal CVD events and 873 (6.0%) individuals with any (fatal or nonfatal) CVD event out of 14,592 participants.

The density plots for diastolic BP, systolic BP, and the BPS are reported in Fig 1. The graphical inspection of these plots confirmed the absence of manipulation around the respective cutoffs, since the density curve runs smoothly through the cutoff (if a manipulation had occurred, we would have seen a jump in the distribution of these variables close to the cutoff). The same result applies to the density plots for the reduced sample for the analysis of the secondary

**Table 1. Descriptive statistics of the analyzed sample.**

| | Full sample | Analytic sample* | | | Difference at the cutoff[†] | |
| --- | --- | --- | --- | --- | --- | --- |
| | | Fatal CVD Event | | | | |
| General characteristics | | Sample | Below cutoff | Above cutoff | Est.[‡] | 95% CI |
| N | 14,592 | 5,556 | 3,142 | 2,414 | | |
| Age (years, SD) | 46 (13) | 48 (13) | 47 (13) | 50 (12) | −1.33 | [−2.88, 0.22] |
| Female | 50% | 39% | 41% | 37% | 1.18 | [0.95, 1.46] |
| High education | 31% | 29% | 29% | 29% | 1.25 | [0.99, 1.57] |
| BMI (kg/m$^2$, SD) | 26 (4) | 27 (4) | 27 (4) | 28 (4) | −0.08 | [−0.50, 0.35] |
| Alcohol (g/day, SD) | 19 (26) | 22 (28) | 21 (25) | 24 (30) | 1.52 | [−1.13, 4.18] |
| Smoking | 29% | 29% | 29% | 28% | 1.24 | [0.97, 1.58] |
| Regular physical activity | 45% | 43% | 45% | 41% | 1.06 | [0.83, 1.34] |
| Previously diagnosed hypertension | 22% | 27% | 23% | 32% | 0.82 | [0.65, 1.05] |
| Systolic BP (mm Hg, SD) | 128 (18) | 135 (8) | 130 (6) | 140 (7) | | |
| Diastolic BP (mm Hg, SD) | 80 (11) | 84 (7) | 82 (6) | 87 (7) | | |
| N events | | 275 (5%) | 122 (4%) | 153 (6%) | | |

BMI, body mass index; BP, blood pressure; CI, confidence interval; CVD, cardiovascular disease; SD, standard deviation.

*Within the optimal respective bandwidths.

[†]Covariate balance tests were computed using a sharp RD model, within the respective optimal bandwidths. See details and graphs in S4 Fig and S4 Table.

[‡]For continuous outcomes, a linear model was used; the resulting estimates are thus expressed as marginal value. For dichotomous outcomes, we used a logistic model; the resulting estimates are thus expressed as odds ratios.

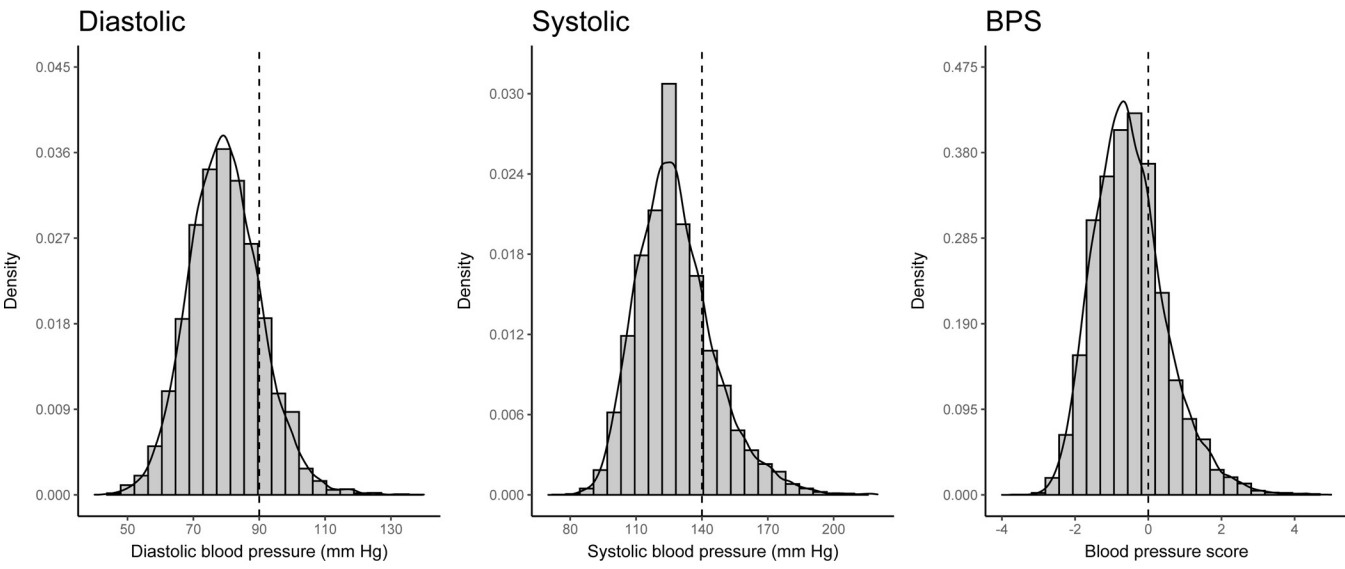

**Fig 1. Density function of diastolic BP, systolic BP, and final BPS.** BP, blood pressure; BPS, blood pressure score.

behavioral outcomes (S3 Fig). Furthermore, the differences at the cutoff reported in Table 1 indicate good covariate balance (for more details, see also S4 Table). The graphical analysis sustained these findings (S4 Fig).

Table 2 and Fig 2A and 2B show the results of the Cox survival model. The RD analysis does not suggest an effect of information treatment on the primary outcomes considered. In fact, both the hazard of fatal CVD events (hazard ratio (HR) = 1.172 [95% confidence interval (CI): 0.725, 1.896]) and the hazard of any CVD event within 16.9 years after information treatment (HR = 1.022 [0.636, 1.641]) were not affected by the information treatment. The stratification for previously diagnosed hypertension at baseline revealed no differential effects. Individuals who stated they were never diagnosed with hypertension previously showed a reduced HR for any CVD event, but the CI is very large (HR = 0.981 [0.609, 1.580]).

Results of secondary analyses using intermediate outcomes revealed no significant changes in systolic or diastolic BP at follow-up after an average follow-up of 8 years (min/max = 6.1/ 10.6 years) (Table 2). Furthermore, we also observed no changes in the share of individuals who were aware of a previous hypertension diagnosis. Behavioral outcomes, including smoking, physical activity, alcohol consumption, and BMI, did not show any significant change following information treatment.

Results for both primary outcomes (Fig 3) and secondary outcomes (S5 and S6 Figs) were robust to the use of original data (instead of imputed data) and different bandwidth choices, which confirmed the direction of the effects and their large CIs.

## Discussion

The analysis does not provide evidence for an effect of a population-based BP screening intervention on long-term CVD mortality and morbidity in Germany. The absence of an effect on long-term outcomes is accompanied by a null effect on secondary or intermediate outcomes. In fact, in the medium term, health behavior as well as BP of individuals remained unchanged.

Several explanations are possible for these null results. First, it might be possible that the "light touch" intervention was not salient enough to induce any change in awareness and behavior. In fact, the diagnosis was communicated within a population-based observational

**Table 2. Sharp RD analysis results.**

| | BW | N | Model | linear Est.[*] | HR/OR[†] | 95% CI |
|---|---|---|---|---|---|---|
| *Primary Outcomes (16.9 years)* | | | | | | |
| Fatal CVD Event | | | | | | |
| Full | 0.575 | 5556 | Cox | - | 1.172 | [0.725, 1.896] |
| Previously diagnosed hypertension (yes) | 0.386 | 1000 | Cox | - | 1.807 | [0.461, 7.073] |
| Previously diagnosed hypertension (no) | 0.654 | 4195 | Cox | - | 1.053 | [0.616, 1.798] |
| Any CVD Event | | | | | | |
| Full | 0.459 | 4368 | Cox | - | 1.022 | [0.636, 1.641] |
| Previously diagnosed hypertension (yes) | 0.457 | 1183 | Cox | - | 1.06 | [0.426, 2.639] |
| Previously diagnosed hypertension (no) | 0.609 | 4124 | Cox | - | 0.981 | [0.609, 1.58] |
| *Intermediate Outcomes (7 years)* | | | | | | |
| BMI | 0.615 | 1879 | Linear | −0.003 | - | [−0.783, 0.777] |
| Alcohol | 0.778 | 2326 | Linear | −0.162 | - | [−3.299, 2.975] |
| Smoking | 0.575 | 1817 | Logit | - | 1.103 | [0.678, 1.797] |
| Regular physical activity | 0.609 | 1858 | Logit | - | 1.17 | [0.812, 1.686] |
| Systolic BP | 0.732 | 2278 | Linear | 0.152 | - | [−2.398, 2.703] |
| Diastolic BP | 0.466 | 1444 | Linear | 0.359 | - | [−1.519, 2.238] |
| Previously diagnosed hypertension | 0.523 | 1515 | Logit | - | 1.146 | [0.76, 1.726] |

BMI, body mass index; BP, blood pressure; BW, optimal bandwidth of the BPS; CI, confidence interval; CVD, cardiovascular disease; HR, hazard ratio; OR, odds ratio; RD, regression discontinuity.

[*]Represents the coefficient on the cutoff variable (tau).

[†]Represents the coefficient on the cutoff variable (tau): HR applies to Cox models; OR applies to Logit models.

study for ethical purposes, without the primary intention of changing behavior. This argument is backed by the fact that individuals above the threshold were not more likely than individuals below the threshold to respond positively to the question "Have you ever been

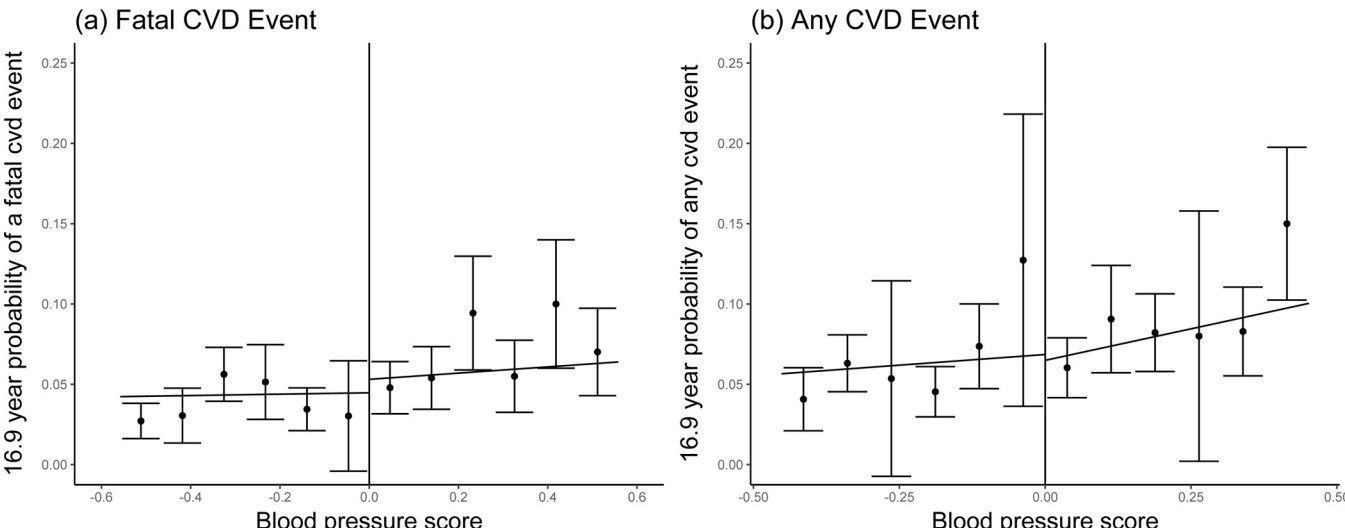

**Fig 2. Graphical representation of the effect at the threshold (with 95% CI). (a and b)** Graphical representation of the effect at the threshold, with linear associations between the assignment variable (BPS) and the respective outcomes for individuals just below and just above the intervention cutoff (BPS = 0) for **(a)** 16.9 years fatal CVD event and **(b)** 16.9 years any CVD event. The 'jump' of the regression line at the threshold represents the intention to treat effect at the threshold. BPS, blood pressure score; CI, confidence interval; CVD, cardiovascular disease.

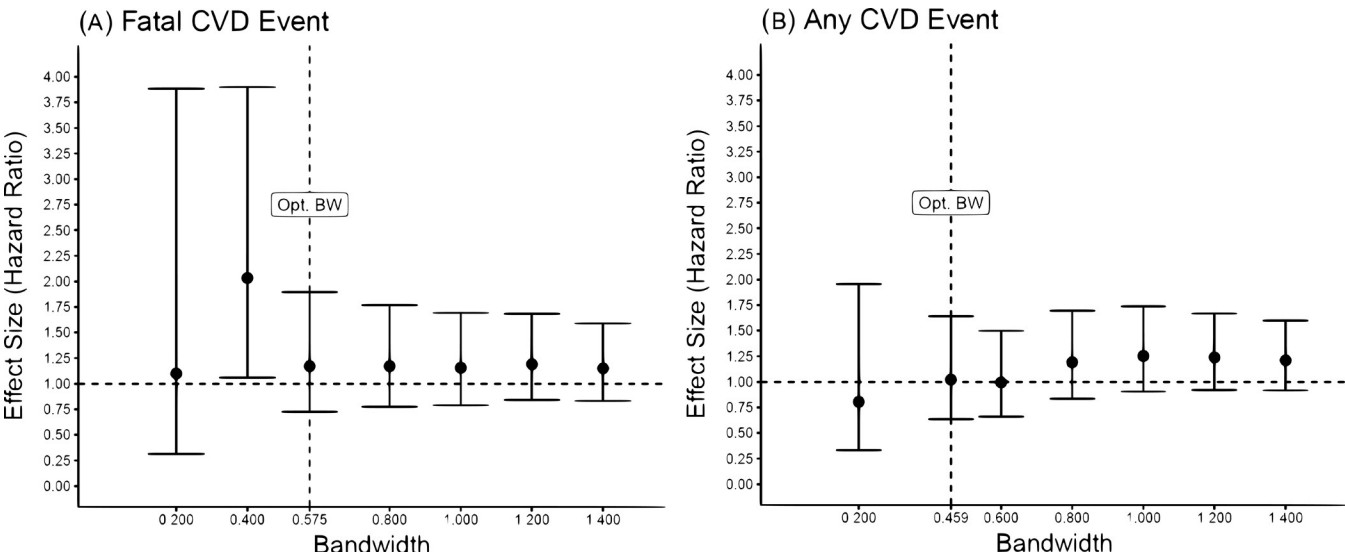

**Fig 3. Graphical representation of the effect at the threshold for different bandwidths.** Graphical representation of sensitivity analysis results on the effect of the intervention at the threshold (and respective 95% CIs) by changing the bandwidth around the cutoff for (Panel A) fatal CVD events results and (Panel B) any CVD event. The dotted line indicates the result with the optimal bandwidth from the main analysis. Other bandwidths indicate larger or narrower bandwidths: A larger bandwidth indicates that we are considering individuals farther away from the cutoff but also a larger number of observations (i.e., up to 1.4 units on the constructed BPS around the cutoff). A narrower bandwidth indicates that we are considering individual closer to the cutoff and therefore less observations (i.e., up to 0.2 units on the constructed BPS around the cutoff). BPS, blood pressure score; CI, confidence interval; CVD, cardiovascular disease.

diagnosed with high blood pressure?" Second, the effect of this "light touch" intervention was not sustainable or large enough to translate into an identifiable effect on CVD morbidity or mortality, within our data. RD design requires, in fact, a large number of observations within the bandwidth around the cutoff, so it is possible that our analysis was not sufficiently powered to detect small changes. Third, it is possible that neither the patient nor their general practitioner (GP) responded to a diagnosis based on the 90/140 mm Hg BP cutoff for hypertension. This might have happened because measurements around this value were not considered worthy of treatment initiation by GPs until changes in the guidelines in the 1990s [35–37], despite evidence being available and despite this threshold was already adopted in official WHO guidelines on hypertension used in the MONICA study in 1985. This fact is also reflected in the low rate of awareness and treatment of hypertension in the older MONICA/KORA studies, compared with more recent follow-ups [38]. Last but not least, the form of the intervention could have led to those just under the threshold altering their behavior or seeking treatment, if, in reading the results letter, they noticed that they were near the threshold. In this way, the behavior of patients and the reaction of physician might not have been very different for those just above and just below the threshold, leading to the observed null result.

The stratification of results for a previously self-reported diagnosis of hypertension showed no differential effects for these two groups for either CVD mortality or any CVD event, including both morbidity and mortality. The effects of this stratification should be interpreted with caution. Whereas the groups of aware versus non-aware individuals might differ substantially in their handling of hypertension information, the self-reported nature of this piece of information may have led to large within-group differences. For example, the group of participants who stated they were never told they had high BP entails both those who really never received such diagnosis as well as individuals who forgot this diagnosis. While this does not prevent both groups to seek medical advice, individuals who were already informed of their condition

and did not seek medical attention nor remembered it are arguably less aware of their health status and therefore less likely to do something against their condition.

## Comparison with previous findings

Our paper adds to the literature investigating the effect of receiving a disease diagnosis on health behavior and health outcomes. Existing evidence has shown that receiving a type 2 diabetes diagnosis [10–13], an overweight diagnosis [13,15–17], an HIV diagnosis [18], or a high cholesterol diagnosis [14] led to mostly modest changes in the relevant health behavior. These effects occurred typically in the short period after the diagnosis but subsequently faded away completely. The majority of these studies used experimental [17] or quasi-experimental methodologies, like RD [12,13,15,16] or instrumental variable (IV) estimation, [18] and thus allowed the identification of a causal effect of diagnosis. As reported by Kim and colleagues [13], one of the reasons why these studies did not find any sustained causal effect of the screening on behavior is that a simple diagnosis might not be effective if it is not combined with further more salient interventions. This result might help to explain the lack of a change in behavior and awareness that we observe in our study, as discussed in detail below.

The current evidence on the effect of hypertension diagnosis differs from the evidence on the effect of the other abovementioned diseases because of the mixed nature of the results hitherto reported. Studies using an RD design in samples from China and South Africa have shown an significant decrease in BP 2 years after diagnosis [4, 5] and a significant improvement in diet 3 to 4 years after diagnosis [6,7]. Conversely, studies on the US population have showed no improvement in dietary patterns after a diagnosis of hypertension within 2 years of diagnosis [8,9]. Therefore, our results appear to support the outcome of those studies carried out in the US, since no effect on behavioral outcomes or long-term morbidity was observed.

However, it must be acknowledged that our study differs significantly from these investigations in several features. First, we considered a medium-term (7 years) time horizon for intermediate outcomes and a long-term time horizon (16.9 years) for CVD mortality and morbidity. In contrast, these previous studies assessed outcomes after 2 to 4 years [4,5].

Second, the studies carried out in the US [8,9] were not based on an experimental or quasi-experimental design but on purely observational methods. Therefore, concerns for residual bias due to unmeasured confounding remain open, limiting the comparability of our results with these previous findings.

Third, another relevant difference between our study and the studies from Chen and colleagues [4] and Sudharsanan and colleagues [5] is the nature and intensity of the information treatment provided. In fact, these two previous studies involved a verbal education on adverse effects of hypertension and on protective lifestyles, followed by treatment encouragement, communicated by field workers orally directly after the screening. It is possible that the "light touch" information treatment considered in this paper, contained in a personalized letter sent to all participants approximately 2 weeks after the screening, was not salient enough to motivate patients to seek medical attention, as noted previously by other authors in a different context [13]. Furthermore, we have no guarantee that patients actually read the letter or showed the results to their physician as suggested. However, the letter outlined the results of a comprehensive medical examination in middle-aged individuals, who participated in a study where most communication between the study center and participants was done by mail. Therefore, we feel it is likely that most participants read it.

Fourth, a further relevant difference between our study and the previously mentioned literature is the context in which the screening was carried out, both in terms of geographical area and time. The papers from Chen and colleagues [4] and Sudharsanan and colleagues [5] were

carried out in middle-income countries, China and South Africa, respectively, whereas our study was carried out in Germany, an HIC with a universal healthcare system. This difference might be relevant to the interpretation of our results, as individuals in Germany with currently unknown hypertension are more likely to be diagnosed with their condition and be treated for it in routine care settings than individuals in LMICs [3]. The effects of a one-time screening might therefore be less pronounced in an HIC compared with an LMIC. Furthermore, the screening investigated here was carried out in four studies conducted between 1985 and 2001. The studies from Chen and Sudharsanan analyzed screening interventions that dated back to 1998 and 2008, respectively. Although the cutoffs of 90/140 mm Hg were already recognized and used in the early cohorts included in this study, it is only recently that such cutoffs have gained importance in the medical community as early intervention points to active treatment of hypertension, since previously a threshold of 95/160 mm Hg was used to diagnose hypertension [35–37]. It is therefore possible that the raised awareness for these lower hypertension thresholds increased the awareness of individuals and the action and treatment by practitioners in more recent years, leading to changes in the short term.

## Strengths and limitations of the study

To our knowledge, this is the first large study looking at the long-term effects of information treatment within a population-based screening. However, using an RD design, we could only estimate the causal effect *at the threshold*, which is not generalizable to the entire population. This precludes the analysis of individuals whose detected BP levels were further right of the threshold, who might have responded differently to the new information than individuals just above the cutoff. Furthermore, we included different surveys over a long time, with changes in standards of care and baseline health. For example, medical guidelines including the underlying thresholds for hypertension threshold changed over the course of our study baseline assessment. This could have led to residual noise, diluting the underlying effect. Additionally, intermediate outcomes were only available for two surveys, limiting the statistical power of secondary analyses. Third, despite assuming that most participants read the results letter, we do not have any proof of this fact. Fourth, we could not analyze the short-term effect of the intervention due to a lack of follow-up studies in the months/years immediately following the baseline studies. This prevents the comparison with other available studies in the literature [4,5] and makes a more detailed analysis of the underlying mechanisms unfeasible. Finally, panel attrition between baseline and the morbidity follow-ups may have prevented the correct identification of self-reported nonfatal strokes and myocardial infarctions. This does not threaten the internal validity of our estimator, but might represent a source of downward bias. Therefore, our estimates should be treated as a lower bound of the underlying effect.

## Implications for research and practice

For clinicians and medical personnel, the results of this study imply that simply informing individuals of their hypertension status might not be enough to change their awareness, to motivate them to seek treatment, and to change behavior in the medium and long term. Further research should investigate how different communication and education strategies might impact awareness and motivation to seek further treatment or adhere to recommendations, as well as potential unintended consequences if, for example, a diagnosis leads to psychological burden [39].

Furthermore, the results from this study combined with previous literature on the effects of disease diagnosis suggest that, if an effect of such screenings and subsequent information treatments exists, it is likely to be a small one, probably limited to a short period of time after

diagnosis. Therefore, in order to detect it, larger datasets with higher statistical power should be pooled and investigated. Furthermore, such pooling would grant the opportunity of investigating also the effect of other thresholds (e.g., 95/160 mm Hg) or other subgroups (previously known versus unknown hypertension) with sufficient statistical power. Lastly, a larger sample would allow the direct consideration of structural differences in the study context, which might considerably alter the final result.

## Conclusions

To the best of our knowledge, this is the first study investigating medium- and long-term impacts of a population-based screening for hypertension on morbidity and mortality outcomes. The results suggest that simply informing patients of their hypertension status is not enough to increase awareness or to improve behavioral intermediate outcomes several years after treatment, nor does it have a positive impact on long-term fatal or nonfatal CVD events. Further research is needed to determine the elements of an effective communication strategy, using a theory-based approach, possibly complementing the communication of the diagnosis with elements and insights from behavioral science [40] and taking into account the trade-off between costly resources and long-lasting effects on health and mortality.

## Supporting information

**S1 Appendix. Collection of drug information.**
(PDF)

**S2 Appendix. Imputation information.**
(PDF)

**S3 Appendix. Supporting information on the intervention.**
(PDF)

**S1 Checklist. TREND checklist.**
(PDF)

**S1 Fig. Flowchart of sample construction for the main analysis.**
(PDF)

**S2 Fig. Flowchart of sample construction for the secondary outcomes analysis.**
(PDF)

**S3 Fig. Density check for the intermediate outcomes sample (at baseline).**
(PDF)

**S4 Fig. Continuity of covariates (at baseline).**
(PDF)

**S5 Fig. Graphical representation of the effect at the threshold for different bandwidths.**
(PDF)

**S6 Fig. Graphical representation of the analysis results with original data.**
(PDF)

**S1 Table. Full descriptive statistics.**
(PDF)

**S2 Table. Imputation data.**
(PDF)

**S3 Table. Description of analytic sample for the outcome "any CVD event."**
(PDF)

**S4 Table. Full information on covariate balance tests.**
(PDF)

## Acknowledgments

The KORA study was initiated and financed by the Helmholtz Zentrum München–German Research Center for Environmental Health, which is funded by the German Federal Ministry of Education and Research (BMBF) and by the State of Bavaria.

## Author Contributions

**Conceptualization:** Sara Pedron, Michael Hanselmann, Jacob Burns, Margit Heier, Michael Laxy.

**Data curation:** Sara Pedron, Michael Hanselmann, Jacob Burns, Alexander Rich, Lars Schwettmann, Michael Laxy.

**Formal analysis:** Sara Pedron, Michael Hanselmann, Jacob Burns, Alexander Rich, Jacob H. Bor, Till Bärnighausen, Michael Laxy.

**Funding acquisition:** Annette Peters.

**Investigation:** Sara Pedron, Michael Hanselmann, Jacob Burns, Alexander Rich, Margit Heier, Lars Schwettmann, Michael Laxy.

**Methodology:** Sara Pedron, Michael Hanselmann, Jacob Burns, Alexander Rich, Jacob H. Bor, Till Bärnighausen, Michael Laxy.

**Project administration:** Sara Pedron, Michael Laxy.

**Resources:** Annette Peters, Margit Heier, Michael Laxy.

**Supervision:** Michael Laxy.

**Writing – original draft:** Sara Pedron, Alexander Rich, Michael Laxy.

**Writing – review & editing:** Sara Pedron, Michael Hanselmann, Jacob Burns, Annette Peters, Margit Heier, Lars Schwettmann, Jacob H. Bor, Till Bärnighausen, Michael Laxy.

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
