## [Editor Report · Decision Letter 0]

10 Aug 2022

Dear Dr Pedron, 

Thank you for submitting your manuscript entitled "The effect of population-based blood pressure screening on long-term cardio-metabolic morbidity and mortality: A regression discontinuity analysis" for consideration by PLOS Medicine.

Your manuscript has now been evaluated by the PLOS Medicine editorial staff and I am writing to let you know that we would like to send your submission out for external peer review.

Please re-submit your manuscript within two working days, i.e. by Aug 12 2022 11:59PM.

Kind regards,

Callam Davidson

Associate Editor

PLOS Medicine

---

## [Decision Letter · Decision Letter 1]

3 Oct 2022

Dear Dr. Pedron,

Thank you very much for submitting your manuscript "The effect of population-based blood pressure screening on long-term cardio-metabolic morbidity and mortality: A regression discontinuity analysis" (PMEDICINE-D-22-02708R1) for consideration at PLOS Medicine. 

Your paper was evaluated by a senior editor and discussed among all the editors here. It was also sent to independent reviewers, including a statistical reviewer. The reviews are appended at the bottom of this email and any accompanying reviewer attachments can be seen via the link below:

[LINK]

In light of these reviews, I am afraid that we will not be able to accept the manuscript for publication in the journal in its current form, but we would like to consider a revised version that addresses the reviewers' and editors' comments. Obviously we cannot make any decision about publication until we have seen the revised manuscript and your response, and we plan to seek re-review by one or more of the reviewers. 

We hope to receive your revised manuscript by Oct 24 2022 11:59PM. Please email us (plosmedicine@plos.org) if you have any questions or concerns.

We look forward to receiving your revised manuscript. 

Sincerely,

Callam Davidson, 

PLOS Medicine

plosmedicine.org

Please include the setting in your title (i.e., “The effect of population-based blood pressure screening on long-term cardio-metabolic morbidity and mortality in Germany: A regression discontinuity analysis”).

Please add this statement to the manuscript's Competing Interests: "TB is editor-in-chief of PLOS Medicine.”

Please structure your abstract using the PLOS Medicine headings (Background, Methods and Findings, Conclusions). Please combine the Methods and Findings sections into one section, “Methods and findings”.

Abstract Background: Provide the context of why the study is important. The final sentence should clearly state the study question. Some of the content in the Introduction should be relocated to the Methods and Findings (e.g., outcomes, length of follow up).

Abstract Methods and Findings:

* Please include the population, years during which the study took place, length of follow up, and main outcome measures.

* Please quantify the main results (with 95% CIs and p values).

Citations should be normal script, located within square brackets, and appear before punctuation (e.g., [1]).

Lines 24-25: Please add ‘to our knowledge’, or similar. 

Please label and describe items in your Supporting Information as outlined here: https://journals.plos.org/plosmedicine/s/supporting-information

Please define the abbreviations in Table S1. 

Please define "lost to follow-up" as used in this study. Other reasons for exclusion should be defined.

Did your study have a prospective protocol or analysis plan? Please state this (either way) early in the Methods section.

Thank you for including a completed TREND checklist. Please use section and paragraph numbers rather than page numbers when completing the checklist, and cite the checklist in your Methods section (as S1 checklist, or similar).

Please remove the ‘Disclosures’ section from the end of the main text and relocate the content regarding ethical approval and consent to the Methods section. The Data Availability information can be removed as this is captured via your Data Availability Statement in the submission form. 

Similarly, the Sources of funding section can be removed and the information instead entered in the Financial Disclosure section of the submission form. 

Figure S1: Please update ‘Gender’ to ‘Sex’.

Line 396: Please temper claims to primacy by adding ‘to our knowledge’, or similar. 

Comments from the reviewers:

Reviewer #1: This study evaluates the effect of blood pressure screening, in the context of a cohort study, on outcomes of blood pressure and fatal or nonfatal CVD events over some 16 years of follow-up. The study addresses a well-known issue and provides new information to inform ongoing debate. The data are analysed using sophisticated methods and the paper is well organised.

1. It is not clear what is the comparator. This is a single group study, but it appears that this intervention did not improve on existing blood pressure care. The paper needs to specify what the latter is. In the UK, general practitioners are encouraged to record the blood pressure of their patients every few years, and a recent cardiovascular health check programme has been found to provide limited additional benefit. Does a similar arrangement exist in Germany? 

2. Use of the term 'screening' can lead to misunderstanding. While public health workers refer to 'popualtion screening', general practitioners and other clinicians use the term 'screening' when they routinely perform tests on specific groups of patients. It would be unfortunate if readers of this paper took from it the message that recording BP is not helpful.

3. The results are generally negative. However, the confidence intervals for the main estimates are very wide leading to questions concerning whether the power of the study is sufficient. Further, 'absence of evidence is not evidence of absence' so it may not be justified to conclude that screening 'did not reduce long-term fatal or nonfatal CVD events' - there is insufficient evidence to draw a conclusion. https://www.bmj.com/content/311/7003/485

4. In the Abstract where it concludes 'More intensive interventions, along with nudging and/or education, may be necessary to affect individual behavior and long-term outcomes.'. This needs to be set in the context of services that already exist (as noted above). No evidence is presented on nudging or education so these ideas need not be mentioned in the conclusion.

5. In the Introduction, this is generally good, but it needs to consider the medical care context as mentioned above. What access did participants have to routine BP care?

6. The Methods are described clearly and in detail. However, at line 127 the term 'bandwidth' is introduced without explanation and the meaning may not be clear. Later it refers to 'optimal bandwidth', with the meaning also unclear.

7. A flowchart detailing the sample selection would be helpful. Where it reads 'The follow-up data for secondary outcomes included 5,130 individuals', please explain why this is reduced as compared to the full sample. 

8. Where it says 'The graphical inspection of these plots confirmed the absence of manipulation around the respective cutoffs.' please explain what is meant by manipulation and why the graphs demonstrate this. 

9. The Figure legends could be more informative. For example, Figure 2 appears to show more than the effect at threshold. Figure 3, meaning of the x axis label of 'bandwidth' may not be clear.

10. In the Discussion, where it says ' The analysis indicated no effect of a population-based blood pressure screening intervention on long-term CVD mortality and morbidity in Germany.' It really means 'no additional effect' over and above existing BP care. But see also point 3 above.

11. The remainder of the discussion and referencing is generally good. It may be worth referring to the distinction of population-wide versus high risk approaches from G Rose, strategy of preventive medicine. It may also be useful to refer to behavioural science literature as finding that 'simply informing patients of their hypertension status is not enough' may not be surprising.

Reviewer #2: See attachment

Michael Dewey

Reviewer #3: In a longitudinal analysis based on 14,592 German adults from four population-based cohort studies free from cardiovascular disease (CVD) and drug treatment at baseline, the Authors investigated the impact of a blood pressure screening and personalized information treatment on CVD morbidity and mortality over a follow-up time of 16.9 years. They concluded no effect of a blood pressure screening intervention on long-term CVD mortality and morbidity and on the medium-term health behaviour or blood pressure intervention. This is a very well written and interesting paper. Statistical approach appears elegant. The discussion and conclusions are well balanced. In the Methods Section, a flow chart of the study design would be of interest. Methods should be improved with more details on information treatment and on the definition of "light touch" intervention. As it is, is it only a mail communication to the patient without direct information to the General Practitioner? If yes, please added a specific statement with explanation (i.e. privacy policy in Germany ecc…). Other studies have found a positive association between correct information about BP to GPs and impact on CVD mortality. Any further direct contact between the medical staff involved in the cohort studies and the participants? Other studies have investigated in longitudinal population-based studies the beneficial impact of observational epidemiological studies on the CVD mortality. On the other hand, some studies have outline that long-term negative psychological effects, as a consequence of the label of HT, exist. Therefore, clinicians might also consider appropriate positive health messages in participants with hypertension (doi:10.1136/openhrt-2015-000341). References should be implemented.

[LINK]

---

## [Decision Letter · Decision Letter 2]

24 Nov 2022

Dear Dr. Pedron,

Thank you very much for re-submitting your manuscript "The effect of population-based blood pressure screening on long-term cardio-metabolic morbidity and mortality in Germany: A regression discontinuity analysis" (PMEDICINE-D-22-02708R2) for review by PLOS Medicine.

I have discussed the paper with my colleagues and the academic editor and it was also seen again by two reviewers. I am pleased to say that provided the remaining editorial and production issues are dealt with we are planning to accept the paper for publication in the journal.

[LINK]

We look forward to receiving the revised manuscript by Dec 01 2022 11:59PM.   

Sincerely,

Callam Davidson, 

Associate Editor 

PLOS Medicine

plosmedicine.org

Requests from Editors:

In the last sentence of the Abstract Methods and Findings section, please describe the main limitations of the study's methodology.

Author Summary: Please include the headline numbers from the study, such as the sample size and key findings.

Line 115: PLOS Medicine does not permit use of footnotes, please relocate this information either to the main text or the Supporting Information.

Please ensure you have cited the relevant Supporting Information correctly in Table 1 legend.

Please ensure consistency in your Supporting Information titles (Figure S6 is inconsistent with the others).

Line 415: Please temper claims to primacy by adding 'to our knowledge'.

Comments from Reviewers:

Reviewer #1: The authors have given an extremely thorough and well considered response to the reviewer comments.

I agree with the authors that P values are not needed in the Abstract. Accordingly, it might be better to replace 'No significant effect' with 'No evidence of effect'.

In the conclusion section of the abstract, where it reads 'Future studies should consider larger datasets'; in terms of the future, would it be worth mentioning the evolving scope of digital health interventions (smarphone apps, wearable devices) that offer increased scope for intervention and evaluation, and whether evaluation of BP in the context of CVD risk (including smoking, cholesterol, diabetes etc) may offer a better approach?

Reviewer #2: The authors have addressed my points. Happy to accept their comment about mice, I was not clear whether that would work but worth a try.

Michael Dewey

[LINK]

---

## [Editor Report · Decision Letter 3]

7 Dec 2022

Dear Dr Pedron, 

On behalf of my colleagues and the Guest Editor, Professor Martin Gulliford, I am pleased to inform you that we have agreed to publish your manuscript "The effect of population-based blood pressure screening on long-term cardio-metabolic morbidity and mortality in Germany: A regression discontinuity analysis" (PMEDICINE-D-22-02708R3) in PLOS Medicine.

To help us extend the reach of your research, please provide any Twitter handle(s) that would be appropriate to tag, including your own, your coauthors’, your institution, funder, or lab. Please respond to this email with any handles you wish to be included when we tweet this paper.

PRESS

Sincerely, 

Callam Davidson 

Associate Editor 

PLOS Medicine